# Skeletal muscle atrophy and myosteatosis are not related to long-term aneurysmal subarachnoid hemorrhage outcome

Yuanyuan Shen[1,2], Stef Levolger[3], Abdallah H. A. Zaid Al-Kaylani[3], Maarten Uyttenboogaart[3,4], Carlina E. van Donkelaar[1], J. Marc C. Van Dijk[1], Alain R. Viddeleer[3], Reinoud P. H. Bokkers[3]*

1 Department of Neurosurgery, University Medical Center Groningen, University of Groningen, Groningen, The Netherlands, 2 Department of Neurosurgery, The First Affiliated Hospital of Zhejiang University School of Medicine, Hangzhou, China, 3 Department of Radiology, Medical Imaging Center, University Medical Center Groningen, University of Groningen, Groningen, The Netherlands, 4 Department of Neurology, University Medical Center Groningen, University of Groningen, Groningen, The Netherlands

* r.p.h.bokkers@umcg.nl

**Data Availability Statement:** Individual data cannot be shared publicly because of General Data Protection Regulation. Aggregated data can be

## Abstract

The prognosis of aneurysmal subarachnoid hemorrhage (aSAH) is highly variable. This study aims to investigate whether skeletal muscle atrophy and myosteatosis are associated with poor outcome after aSAH. In this study, a cohort of 293 consecutive aSAH-patients admitted during a 4-year period was retrospectively analyzed. Cross-sectional muscle measurements were obtained at the level of the third cervical vertebra. Muscle atrophy was defined by a sex-specific cutoff value. Myosteatosis was defined by a BMI-specific cutoff value. Poor neurological outcome was defined as modified Rankin Scale 4–6 at 2 and 6-month follow-up. Patient survival state was checked until January 2021. Generalized estimating equation was performed to assess the effect of muscle atrophy / myosteatosis on poor neurological outcome after aSAH. Cox regression was performed to analyze the impact of muscle atrophy and myosteatosis on overall survival. The study found that myosteatosis was associated with poor neurological condition (WFNS 4–5) at admission after adjusting for covariates (odds ratio [OR] 2.01; 95%CI 1.05,3.83; P = .03). It was not associated with overall survival (P = .89) or with poor neurological outcomes (P = .18) when adjusted for other prognostic markers. Muscle atrophy was not associated with overall survival (P = .58) or neurological outcome (P = .32) after aSAH. In conclusion, myosteatosis was found to be associated with poor physical condition directly after onset of aSAH. Skeletal muscle atrophy and myosteatosis were however irrelevant to outcome in the Western-European aSAH patient. Future studies are needed to validate these finding.

## Introduction

Aneurysmal subarachnoid hemorrhage (aSAH) is an acute neurological emergency with a global incidence of 6.1 per 100,000 person-years [1]. Its mortality ranges from 8.3% up to

made available upon request for researchers who meet the criteria for access to confidential data. The Data Protection Officers of the University Medical Centre Groningen can be contacted at privacy@umcg.nl.

**Funding:** YS receives financial support from China Scholarship Council (CSC, File No. 201706320024, https://www.csc.edu.cn/) The funders had no role in study design, data collection and analysis, decision to publish, or preparation of the manuscript.

**Competing interests:** The authors have declared that no competing interests exist.

66.7% [2]. One-third of the survivors have motor/language impairments or disabilities in daily living activities at five-year follow-up [3, 4]. The most important predictive factors for clinical outcome are level of consciousness and neurological deficits at admission, patient age, and the amount of blood on the initial computed tomography (CT) [5].

Recently, temporal muscle thickness and area measurements delineated on CT have been associated with poor clinical outcome in elderly aSAH patients in a Japanese population [6]. Although temporal muscle measurements are not directly linked to sarcopenia, the findings indicate that muscle-biomarkers may be used as a predictor for aSAH outcome [7]. Sarcopenia is recognized as a progressive decline in muscle mass and strength that occurs across a lifetime [7]. In overlap with frailty, it can be considered an age-related decline in physiological reserve [8]. Sarcopenia predisposes patients to a wide range of negative health-related events and worse outcome in various diseases, e.g. lung cancer, chronic obstructive pulmonary disease (COPD), cachexia or chronic heart disease [9–12] and in patients who underwent major surgery, including abdominal aortic aneurysm repair [13, 14]. Another biomarker of muscle change is myosteatosis, which is an indicator of muscle quality. It is defined as the accumulation of intramuscular and intermuscular fat [15, 16]. Neither of these muscle alterations have an identical association with increased morbidity and mortality. In hospitalized geriatric patients, myosteatosis is associated with high mortality only in male patients, while sarcopenia was a risk factor for overall death among the whole geriatric cohort [17–20].

The aim of this study was to investigate the association between skeletal muscle atrophy, myosteatosis, and poor outcome after aneurysmatic subarachnoid hemorrhage.

## Methods

This is a retrospective cohort study of aSAH-patients with a CT-scan that covered the third cervical vertebral body. All patients were consecutively treated at the University Medical Center Groningen (UMCG) between January 2013 and December 2017 and had a follow-up period of at least 3 years. The UMCG is a regional comprehensive neurovascular center in the North of The Netherlands. Patients with a history of muscular disease or lacking height and weight records were excluded.

The study was approved by the UMCG ethics review board (METc 2019/505). According to Dutch regulations and General Data Protection Regulation, no informed consent was required due to the retrospective and observational nature of this study. Access to the national Personal Records Database (BRP) in order to determine the survival status was approved by the National Service for Identity Data (RvIG).

### Patient characteristics

Patients' clinical data were extracted from the electronic medical records, including sex; age at ictus; height and weight (at admission or within 3 days); history of SAH and myocardial infarction; presence of hypertension (systolic blood pressure >140 mmHg or diastolic blood pressure >90 mmHg or controlled using antihypertensive drugs). The World Federation of Neurosurgical Societies (WFNS) grading system was used to assess neurological condition at admission [21]; if a resuscitation was performed before aneurysm reparation, the WFNS-grade after resuscitation within 12 hours was used [22]. WFNS 4–5 was considered a poor neurological condition. The severity of hemorrhage was assessed on the initial CT using the modified Fisher scale. Grade 3–4 was categorized as high mFisher. The size of the aneurysm was defined large if greater than 10 mm.

## Image acquisition and muscle measurements

All patients had an initial CT-angiogram of the head and neck as a part of the standard clinical care. CT-imaging was performed on a Siemens SOMATOM Definition Edge or SOMATOM Sensation (Siemens Medical, Erlangen, Germany), using a 512x512 matrix, soft-tissue reconstruction kernel and a slice thickness of 3–5 mm. All scans were performed with intravenous contrast in the arterial phase. CT-images were retrieved from the hospital's picture archiving and communications system (PACS) and saved in Digital Imaging and Communications in Medicine (DICOM) format for further analysis.

Cross-sectional muscle measurement was obtained at the level of the third cervical vertebra (C3), on the slice where both superior articular processes were shown. The paravertebral muscle (PVM) included rotator cervicis, levator scapulae, longissimus capitis, interspinales cervicis, semispinalis cervicis, semispinalis capitis, splenius capitis, and trapezius muscles. Bilateral sternocleidomastoid muscles (SCM) and PVM were considered the region of interest (ROI) in this study (Fig 1).

In-house developed software (SarcoMeas Neck 0.34; UMCG, Groningen, The Netherlands) was used to perform the skeletal muscle measurements. With this software, the neck muscles at the level of the third cervical vertebra were manually delineated by one investigator (AK) based on radiodensity as expressed in Hounsfield Units (HU). The target muscles are illustrated in Fig 1. Within these contours, voxels with a radiodensity from -29 to 150 HU were identified as skeletal muscle. The Skeletal Muscle Area (SMA) was determined by calculating the volume of all muscle voxels. The SMA was then corrected for patient length, by dividing the muscle area by the squared patient length, resulting in the Skeletal Muscle Index (SMI), expressed in $cm^2/m^2$. Skeletal Muscle Density (SMD) was defined as the mean density of all muscle voxels within the drawn contours.

To assess interobserver variability, all measurements were independently repeated by a second investigator (AV). Moreover, 50 random cases were remeasured by the first investigator (AK) after two years, for assessment of intraobserver agreement.

## Treatment and follow-up

All aSAH-patients were treated following a standardized multidisciplinary protocol [22]. Ruptured aneurysms in patients with good neurological grades (WFNS 1–3) were clipped or coiled within 72 hours after the ictus. In patients with poor condition (WFNS 4–5), neurological resuscitation in the intensive care unit was instituted and urgent interventions (e.g., CSF drainage or hematoma evacuation) were performed within 24 hours.

## Outcome parameters

The primary outcome was mortality. The national civil registry was consulted for long-term survival data on January 20th, 2021. Survival time was defined as the period between aSAH and death, or ultimately January 20th 2021. The secondary outcome measure was neurological function (modified Rankin Scale—mRS [23]), graded at 2 and 6 months follow-up with a questionnaire. An mRS 4–6 was considered poor.

## Statistical analysis

Difference in baseline characteristics between groups was tested using a t-test for continuous variables if normally distributed, Mann-Whitney U test in case of non-normal distribution, and a chi-square test or Fisher exact test for categorical variables [24].

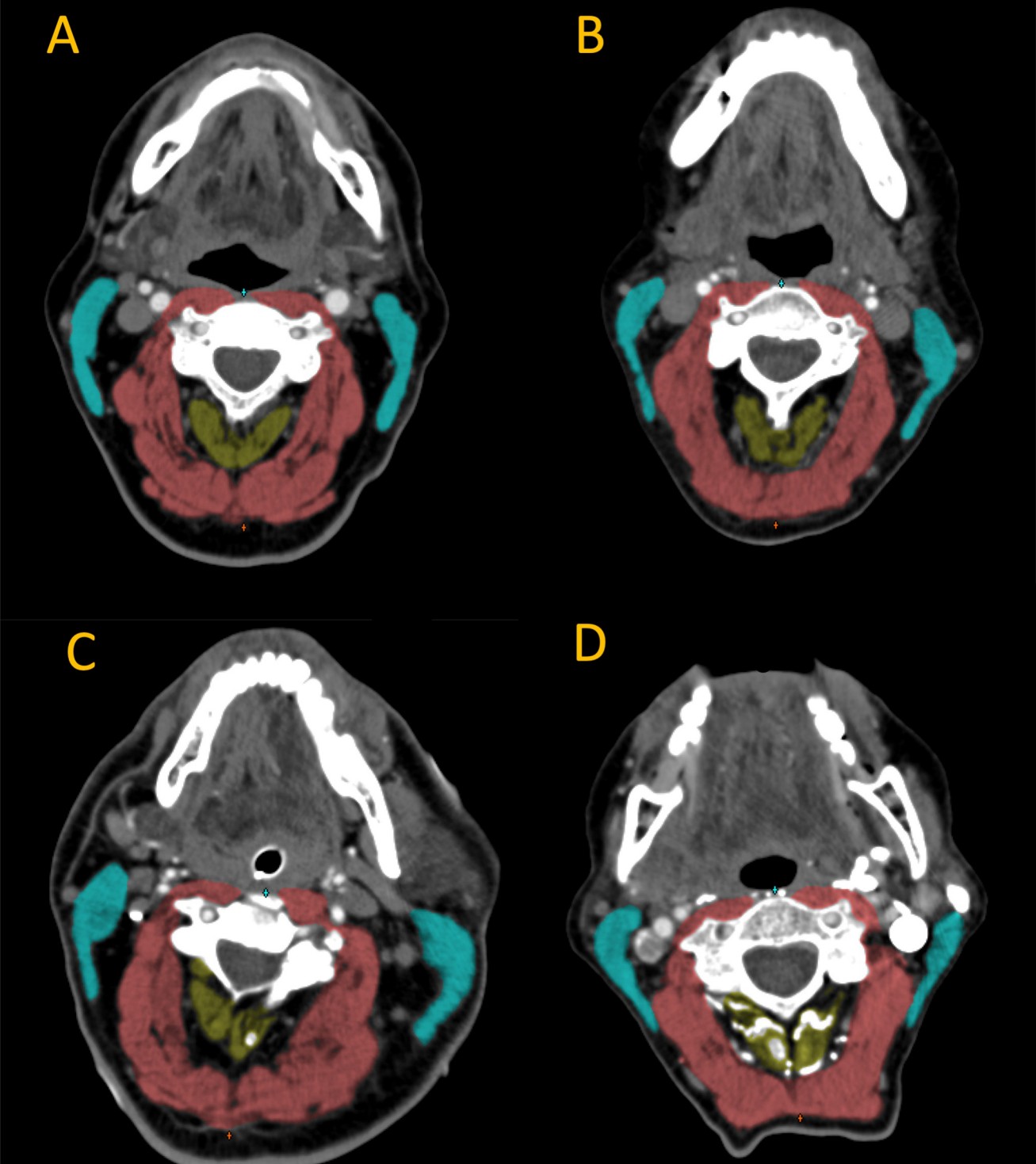

**Fig 1. Schematic diagram of cross-sectional muscle measured at the level of third cervical vertebra.** Muscles of interest are colored as **cyan** for sternocleidomastoid muscles; **yellow** for interspinales cervicis; **red** for rotator cervicis, levator scapulae, longissimus capitis, semispinalis cervicis, semispinalis capitis, splenius capitis, and trapezius muscles. **A** cross-sectional image from a patient without skeletal muscle atrophy or myosteatosis (female, BMI 27.5, SMI 13.2, mean HU 43.8); **B** a patient with both muscle atrophy and myosteatosis (male, BMI 21.5, SMI 12.1, mean HU 40.8); **C** a patient with myosteatosis but no muscle atrophy (male, BMI 26.6, SMI 14.5 mean HU 36.8); **D** a patient with muscle atrophy but non-myosteatosis (female, BMI 20.3, SMI 10.8, mean HU 48.0). BMI: body mass index; SMI: skeletal muscle index (SMA / patient height2); HU: Hounsfield Units.

The presence of myosteatosis was determined by means of applying a threshold. From previous studies it is known that the threshold of myosteatosis varies significantly based on the BMI of the subject [9, 25]. Therefore, we established BMI-specific cutoff points of mean HU value for myosteatosis by optimum stratification. Optimal stratification is a strategy, based on log-rank statistics to determine at which cutoff value the most significant difference would occur with regard to a binary outcome or a time-to-event outcome, such as death in this study [26]. A two-step procedure was performed to determine the thresholds: First, all patients were split into two subgroups based on the value of BMI $25kg/m^2$ according to the WHO definition of overweight [27]; Secondly, within each subgroup, the mean HU value with the most significant P value for overall mortality by log rank statistics was selected as the cutoff point of the BMI-specific group.

Muscle mass in males is furthermore known to differ significantly from that in females [28]. To determine the presence of muscle atrophy, we corrected for this by applying a sex-specific threshold for muscle atrophy in accordance to previous studies [25]. An attempt was made to determine the SMI threshold following the two-step procedure as stated above, however a threshold was not found within this cohort. Previously reported gender-specific cutoff points for sarcopenia were therefore used ($7.8 cm^2/m^2$ for female, and $12.1 cm^2/m^2$ for male) [29].

Cox proportional hazards model was used to assess the correlation between muscle alternations and mortality. Proportional Hazards Assumption was not significant. Covariables were selected based on: 1. Variables that were significantly different between groups; 2. Variables that were associated with death at univariable analysis; 3. Variables that had more than 10 events (death). The two variables were binarized as good/poor WFNS and low/high mFisher. As an age of 70 or older is an independent risk factor for poor neurological outcome 2 month after aneurysmal SAH, age was dichotomized at 70 years [30]. A forward likelihood ratio method was used for model fitting. All covariables were checked for interaction and confounding. Generalized estimating equation was used to assess the effect of skeletal muscle atrophy and myosteatosis on neurological outcome. A two-tail p-value < .05 was considered statistically significant. Statistical analysis was performed with SPSS version 25.0 (IBM, Armonk, NY).

## Results

During the inclusion period, 518 consecutive patients were diagnosed and treated for aSAH, of which 293 patients were eligible for skeletal muscle analysis. Fig 2 depicts the enrollment process; 58 patients were excluded because the CT-images were insufficient (cross-section at C3 level was not complete); 167 patients were excluded for unavailable patient height; 2 more patients were excluded from the myosteatosis analyses as their weight was missing. The two-month follow up interview and quality of life assessment was incomplete in 7 patients, as well as in 57 patients at six-month follow up. The survival-state was determined on January 20[th], 2021 for all 293 patients.

Baseline patient characteristics are reported in Table 1. In 293 (99%) subjects the slice thickness was 3mm and in 4 (1%) patients it was 5mm. The median follow-up time was 58.0 months (interquartile range 33.0 month). Sixty-eight patients died during follow-up. Continuous variables were all normally distributed.

Both interobserver and intraobserver levels of agreement for SMA and mean HU value were found to be excellent (Table 2).

### Skeletal muscle atrophy

Skeletal muscle atrophy was identified in 14 of 293 (4.8%) patients. Compared to the patients without muscle atrophy, fewer female patients were present in the muscle atrophic cohort

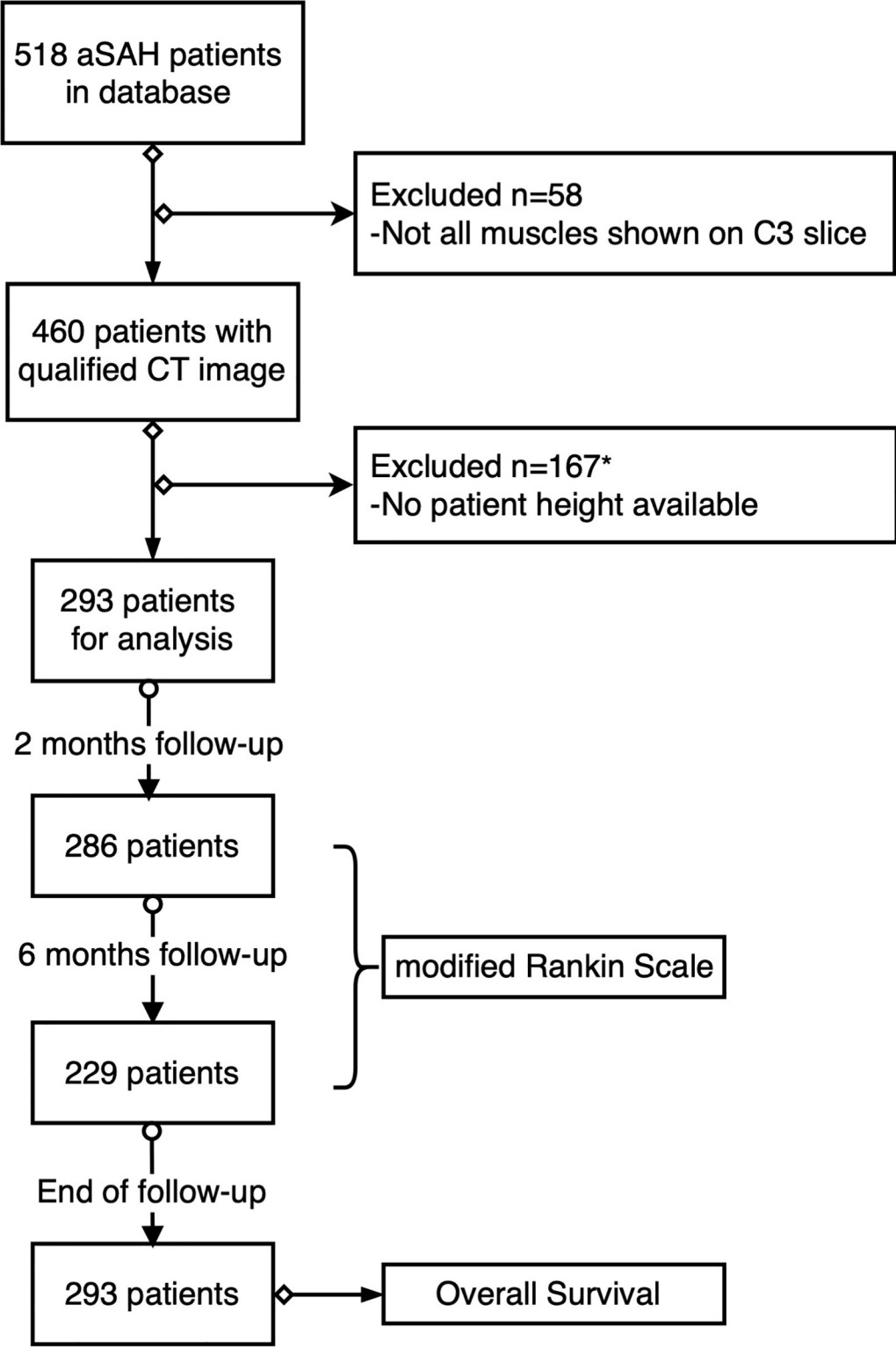

**Fig 2. Study flowchart.** Outcome of this study included two parts: survival state and quality of life represented by modified Rankin Scale. Modified Rankin Scale is assessed at 2 months and 6 months follow-up, survival state of all patients was required on January 20th, 2021. * The weight of two more patients were unavailable for defining myosteatosis.

**Table 1. Patient characteristics.**

| | | Myosteatosis | | | Skeletal Muscle atrophy | | |
|---|---|---|---|---|---|---|---|
| | | No (n = 160) | Yes (n = 145) | *P* value | No (n = 279) | Yes (n = 14) | *P* value |
| **Female** | | 96(60.0) | 108(74.5) | **.01** | 193(69.2) | 1(7.1) | < **.001** |
| **Age years** | | 53.6(13.6) | 62.9(11.4) | .35 | 57.89(13.2) | 59.0(13.1) | .76 |
| **Age >70y** | | 15(9.4) | 45(31.0) | < **.001** | 53(19.0) | 1(7.1) | .48 |
| **BMI kg/cm²** | | 25.3(4.3) | 27.2(5.7) | **.03** | 26.4(5.1) | 22.9(2.7) | **.01** |
| **BMI<25kg/cm²** | | 81(50.6) | 62(42.8) | .21 | 126(45.5) | 12(85.7) | < **.01** |
| **Pre SAH** | | 4(2.5) | 6(4.1) | .53 | 9(3.2) | 1(7.1) | .39 |
| **Hypertension** | | 42(26.3) | 59(40.7) | **.01** | 93(33.3) | 3(21.4) | .56 |
| **MI** | | 7(4.4) | 5(3.4) | .77 | 10(3.6) | 1(7.1) | .42 |
| **Smoking** | | 11(16.9) | 13(22) | .50 | 21(18.4) | 1(20) | 1.00 |
| **Alcohol** | | 3(4.6) | 4(6.8) | .71 | 7(6.1) | 0 | 1.00 |
| **WFNS** | 1 | 87(54.4) | 46(34.8) | < **.01** | 123(44.1) | 10(71.4) | .12 |
| | 2 | 33(20.6) | 33(25.0) | | 66(23.7) | 0 | |
| | 3 | 9(5.7) | 6(4.5) | | 15(5.4) | 0 | |
| | 4 | 15(9.4) | 24(18.2) | | 38(13.6) | 1(7.1) | |
| | 5 | 15(9.4) | 23(17.4) | | 37(13.3) | 3(21.4) | |
| **Poor WFNS** | | 30(18.9) | 47(35.6) | < **.01** | 75(26.9) | 4(28.6) | 1.00 |
| **mFisher** | 0 | 12(7.5) | 0 | < **.001** | 11(3.9) | 1(7.1) | .05 |
| | 1 | 41(25.8) | 16(11) | | 51(18.3) | 7(50) | |
| | 2 | 33(20.8) | 40(27.6) | | 69(24.7) | 1(7.1) | |
| | 3 | 23(14.5) | 15(10.3) | | 37(13.3) | 1(7.1) | |
| | 4 | 50(31.4) | 74(51) | | 111(39.8) | 4(28.6) | |
| **High mFisher** | | 73(45.9) | 79(59.8) | **.02** | 148(53) | 5(35.7) | .21 |
| **Large Aneurysm** | | 34(22.5) | 32(23.7) | .89 | 61(23.3) | 2(15.4) | .74 |
| **SMI cm²/m²** | male n = 99 | 14.8(2.7) | 14.7(2.5) | .78 | 15.4(2.2) | 10.7(1.1) | < **.001** |
| | female n = 194 | 12.0(2.0) | 7.4 | **.03** | 11.8(2.0) | 12.1(2.1) | .26 |
| **Skeletal Muscle atrophy** | | 10(6.3) | 4(3.0) | .27 | | | |
| **mean HU** | BMI<25 n = 143 | 48.7(7.2) | 36.6(4.3) | < **.001** | 43.5(8.5) | 44.5(9.0) | .72 |
| | BMI≥25 n = 162 | 44.1(4.1) | 34.3(4.2) | < **.001** | 39.3(6.4) | 43.0(5.0) | .41 |
| **Myosteatosis** | | | | | 128(46.2) | 4(28.6) | .27 |

All continuous variables were normally distributed, reported as mean and standard deviation. Categorical variables were presented as case number and percentage of the group. BMI, Body mass index; pre SAH, history of Subarachnoid hemorrhage; MI, myocardial infarction; Poor WFNS, World Federation of Neurosurgical Societies grading 4 or 5; High mFisher, modified Fisher scale 3 or 4; SMI, skeletal muscle index; Large aneurysm, aneurysm size larger than 10 mm. HU, Hounsfield unit.

(7.1% vs 69.2%, P< .001). Also, muscle atrophic patients had a lower mean BMI (22.93 ± 2.70 vs 26.35 ± 5.12, P = .01). There was no statistical difference in age, smoking, alcohol, aneurysm size, WFNS, or mFisher scale. Overall survival was equal in patients with and without muscle atrophy (P = .56, Table 3).

**Table 2. Interobserver and intraobserver levels of agreement for SMI and mean HU value.**

| Agreement* | Skeletal Muscle Area | Mean HU |
|---|---|---|
| Interobserver | 0.988 (95%CI 0.984, 0.991) | 0.995 (95%CI 0.990, 0.997) |
| Intraobserver | 0.975 (95%CI 0.884, 0.991) | 0.908 (95%CI 0.832, 0.949) |

*Intraclass correlation coefficient, absolute agreement, two-way random, average measures.

**Table 3. Cox regression of overall survival.**

| Variable | Univariable analysis | | | Multivariable analysis | | |
|---|---|---|---|---|---|---|
| | HR | 95% CI | P value | HR | 95% CI | P value |
| Female | 1.14 | 0.80–1.60 | 0.46 | | | |
| Age | 1.04 | 1.02–1.05 | **<0.001** | 1.06 | 1.04–1.09 | **<0.001** |
| BMI | 1.03 | 0.98–1.07 | 0.27 | | | |
| Previous SAH | 1.66 | 0.78–3.56 | 0.19 | | | |
| Hypertension | 1.45 | 1.02–2.06 | **0.04** | 0.66 | 0.38–1.15 | 0.15 |
| Smoking | 0.75 | 0.34–1.65 | 0.47 | | | |
| Alcohol | 1.53 | 0.61–3.84 | 0.36 | | | |
| Poor WFNS | 5.65 | 3.99–8.00 | **<0.001** | 2.75 | 1.52–4.98 | **0.001** |
| High mFisher | 3.60 | 2.45–5.30 | **<0.001** | 1.40 | 0.75–2.61 | 0.29 |
| Large aneurysm | 2.15 | 1.48–3.12 | **<0.001** | 1.42 | 0.82–2.48 | 0.22 |
| Sarcopenia | 1.32 | 0.48–3.64 | 0.58 | | | |
| Myosteatosis | 1.97 | 1.20–3.22 | **0.01** | 1.04 | 0.60–1.81 | 0.89 |

Covariates in multivariable analysis: age, present of hypertension, large aneurysm, high mFisher, poor WFNS, myosteatosis. HR, Hazard ratio; BMI, Body mass index; SAH, Subarachnoid hemorrhage; MI, myocardial infarction; Poor WFNS, World Federation of Neurosurgical Societies grading 4 or 5; High mFisher, modified Fisher scale 3 or 4; Large aneurysm, aneurysm size larger than 10 mm.

Survival at month 12, 36, and 60 was 78.6%, 71.4%, 71.4% for muscle atrophic patients, and 84.9%, 81.7%, 78.5% for patients without muscle atrophy (Fig 3). Skeletal muscle atrophy was not associated with neurological outcome (P = .32, Table 4).

## Myosteatosis

Myosteatosis is the accumulation of intra- and inter-muscle adipose tissue, which leads to the decreasing of radiodensity expressed in HU value. BMI-specific mean HU cutoff values (optimum stratification) were 39.6 for BMI < 25 kg/m$^2$, and 42.1 for BMI ≥ 25 kg/m$^2$. Non-myosteatotic and myosteatotic patients had overlapping age-groups (53.6 ± 13.6 vs 62.9 ± 11.4, P = .35), but patients in the myosteatotic group were more likely to be above the age of 70 (31.0% vs 9.4%, P< .001). Furthermore, myosteatotic patients showed a higher female preponderance (74.5% vs 60.0%, P = .01); higher BMI (27.2 ± 5.7 vs 25.3 ± 4.3, P = .03); higher prevalence of arterial hypertension (40.7% vs 26.3%, P = .01); higher frequency of poor WFNS grade (35.6% vs 18.9%, P< .001); and higher mFisher score (59.8% vs 45.9%, P = .02). No difference was found in aneurysm size, history of previous SAH, smoking, or alcohol consumption. Considering age, presence of hypertension, aneurysm size, and mFisher scale as covariates, the adjusted OR of myosteatosis on poor WFNS grade was 2.01 (95%CI 1.05, 3.83; P = .03). The correlation coefficient of mean HU value versus WFNS grade and mFisher scale was 0.16 and 0.15 respectively (Table 5).

Survival at month 12, 36, 60, and 84 was 91.7%, 77.3%, 72.0%, and 65.8% for myosteatotic patients and 92.5%, 85.5%, 85.5% and 82.8% for non-myosteatotic patients respectively (Fig 3). All-cause mortality of myosteatotic patients was increased in univariable analysis (P = .01, Table 2), but not in multivariable analysis (P = .89). Also, more myosteatotic patients had a poor neurological condition at two month (38.0% vs 18.7%, P< .001) and at six months follow up (30.8% vs 12.5%, P< .01). Considering the effect over time, the association between myosteatosis and poor neurological outcomes remained with OR 2.83 (95%CI 1.67,4.79; P< .001). However, when adjusted for covariates, the statistically significant effect disappeared (Table 4).

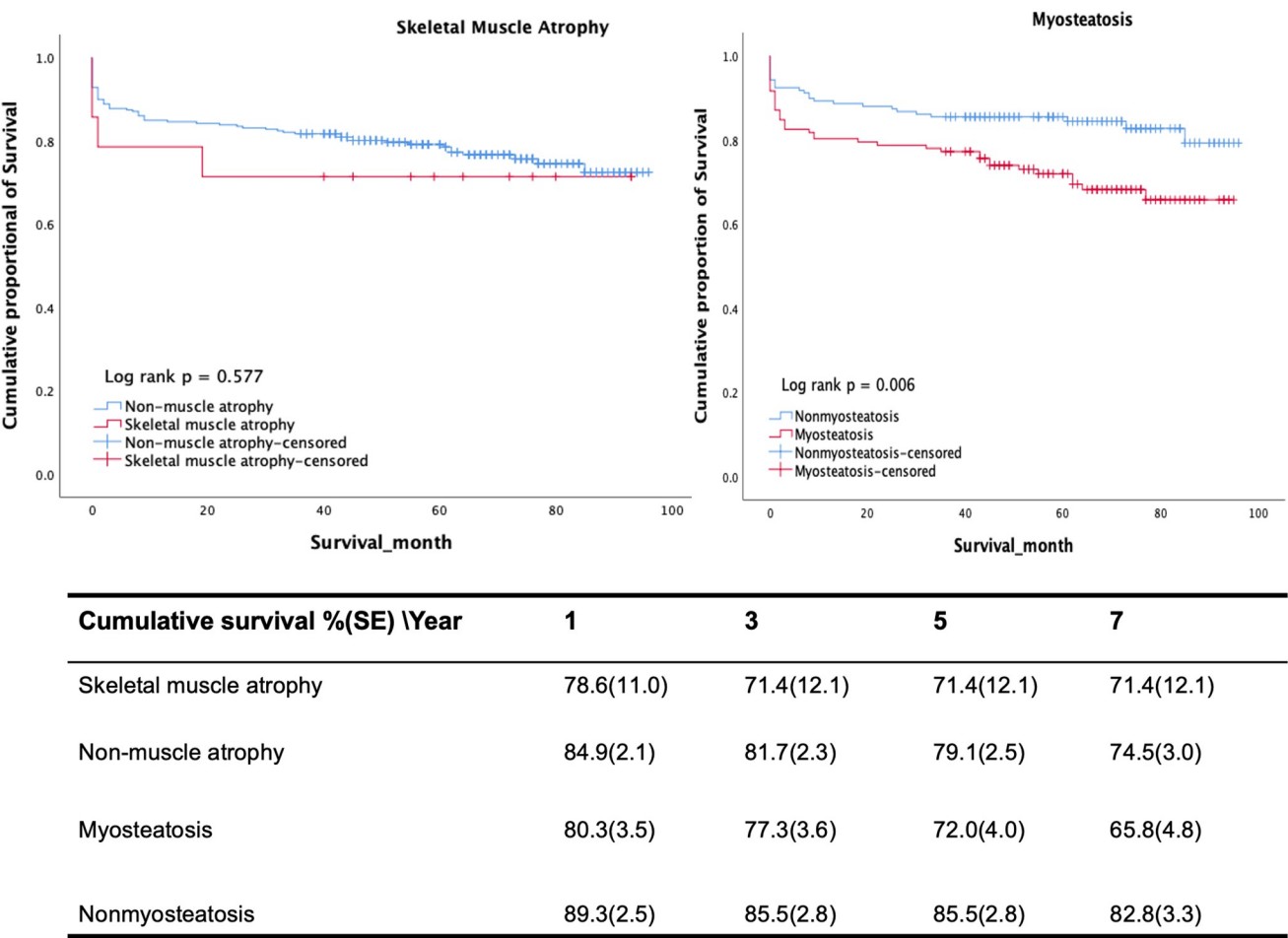

| Cumulative survival %(SE) \Year | 1 | 3 | 5 | 7 |
|---|---|---|---|---|
| Skeletal muscle atrophy | 78.6(11.0) | 71.4(12.1) | 71.4(12.1) | 71.4(12.1) |
| Non-muscle atrophy | 84.9(2.1) | 81.7(2.3) | 79.1(2.5) | 74.5(3.0) |
| Myosteatosis | 80.3(3.5) | 77.3(3.6) | 72.0(4.0) | 65.8(4.8) |
| Nonmyosteatosis | 89.3(2.5) | 85.5(2.8) | 85.5(2.8) | 82.8(3.3) |

**Fig 3. Kaplan-Meier survival curves of patients with skeletal muscle atrophy and myosteatosis.**

**Table 4. Effects of muscle alterations on neurological outcomes.**

| | | | 2 months[*] | 6 months[*] | Effect over time[**] |
|---|---|---|---|---|---|
| **Muscle atrophy** | Univariable analysis | OR (95% CI) | 1.31 (0.35, 4.89) | 2.59 (0.32, 20.98) | 1.94 (0.52, 7.27) |
| | | P value | .69 | .70 | .32 |
| **Myosteatosis** | Univariable analysis | OR (95% CI) | 2.66 (1.55, 4.56) | 3.12 (1.58, 6.16) | 2.83(1.67, 4.79) |
| | | P value | < **.001** | < **.01** | < **.001** |
| | Multivariable analysis | OR (95% CI) | 1.64 (0.78, 3.46) | 1.49 (0.60, 3.69) | 0.63 (0.31, 1.28) |
| | | P value | .19 | .39 | .20 |

Covariates in multivariable analysis: age, present of hypertension, large aneurysm, high mFisher, poor WFNS, myosteatosis.

[*] Binary logistic regression

[**] Generalized estimating equation.

**Table 5. Effect of myosteatosis on mFisher scale and WFNS grade.**

| | | | High mFisher | Poor WFNS | mFisher | WFNS |
|---|---|---|---|---|---|---|
| Myosteatosis* | Univariable analysis | OR (95% CI) | 1.76 (1.10, 2.80) | 2.38 (1.39, 4.05) | | |
| | | *P* value | **.02** | **< .01** | | |
| | Multivariable analysis | OR (95%CI) | 1.20 (0.69, 2.08) | 2.01 (1.05, 3.83) | | |
| | | *P* value | .51 | **.03** | | |
| mean HU** | Correlation Coefficient | | 0.15 | 0.16 | 0.20 | 0.15 |
| | *P* value | | < .001 | < .001 | < .001 | < .001 |

OR, Odds Ratio; CI, Confidence Interval; Poor WFNS, World Federation of Neurosurgical Societies grading 4 or 5; High mFisher, modified Fisher scale 3 or 4.

*Binary logistic regression, covariates in multivariable analysis: age, present of hypertension, large aneurysm, and high mFisher scale/ poor WFNS.

** Kendall's tau_b correlation coefficient.

## Discussion

The current study showed no association between skeletal muscle atrophy, myosteatosis and mortality or neurological outcome. Myosteatosis was correlated with the severity of aSAH at onset. To the author's knowledge, the impact of muscle atrophy and myosteatosis on all-cause mortality and neurological outcome in aSAH patients in a western population has not yet been reported.

Our findings contradict the association between muscle atrophy and poor mRS as reported in studies with a Japanese aSAH cohort [6, 31]. In that study, temporal muscle area and thickness were measured as indicators for sarcopenia in contrast to this study. However, since the temporal muscle is a minor muscle, its reduction may be less representative of overall skeletal muscle [32]. Measurement at the abdominal level can represent total body skeletal muscle and adipose tissue volumes [33]. Skeletal muscle mass and index (SMM/SMI) at the third lumbar vertebra (L3) is prevalently used to define sarcopenia. Since SMI at the 3rd cervical vertebra (C3) is strongly associated with the L3 level [29, 34], the C3 level was used for muscle measurement in this study. Additional to this difference in methodology, population-based differences are also important. Aneurysm rupture risk is known to be higher in Japanese than in the western population (disregarding Finland) [35]. As such, it should be pointed out that, different from that Japanese cohort, our study was based on a Dutch population.

Sarcopenia and myosteatosis have been considered as negative prognostic indexes of varied chronic or consumptive conditions, (e.g. COPD, chronic heart failure, liver cirrhosis, cancers [36–39]), as well as recovery ability markers from major surgery (e.g. laryngectomy, radiotherapy, transcatheter aortic valve replacement, and abdominal aortic aneurysm repair [13, 14, 40–42]). Our primary hypothesis that skeletal muscle atrophy and myosteatosis are negative prognostic markers of aSAH is rejected. However, apart from the irrelevance to aSAH outcome, myosteatosis and mean HU value was associated with poor WFNS and high mFisher at baseline. The mFisher scale classifies the hemorrhage volume, while WFNS depicts the physical reaction to the hemorrhage. Both reflect in different ways the severity of aSAH and are nowadays the most predictive outcome predictors of aSAH [5].

Similar to undergoing surgery, aSAH induces an acute stress response. This induced stress syndrome and accompanying hypercatabolic state has been thoroughly described post-surgery and in trauma patients [43, 44]. The protein synthesis rate decreases not only after major surgery, but also after relatively minor surgery [45]. In patients with severe trauma but free from infection, the mortality is higher in a group with lower synthetic rate of proteins than those with accelerated synthetic rate of proteins [46]. Fat-free muscle mass reflects the reserve

capacity of muscle proteins and is associated with the occurrence of complications in hospitalized patients and among patients with COPD [47–49]. A study on skeletal muscle mass and post-operation conditions among patients undergoing cardiac surgery suggested that preoperative fat-free muscle mass indicates the ability to cope with operative stress, reflected by post-operation complications [50]. In aSAH patients a comparable hypercatabolic state is observed [51]. Furthermore, daily systemic energy expenditure is increased in the acute phase [52]. And systemic inflammation occurs, including upregulation of cytokines such as IL-6 [53], which particularly in illness is associated with atrophy and muscle wasting [54]. Consequently, despite the negative results found in our study in relation to skeletal muscle atrophy, myosteatosis and overall survival, the differences in muscle quantity and quality observed to be associated with physical condition may reflect low physical reserve. In line with this, a recent study showed high protein intake after SAH actually reduces the rates of temporal muscle atrophy [55]. Taking all into consideration, it is of value to investigate whether early aggressive anticatabolic treatment or nutritional supportive treatment can improve the physical performance outcome of aSAH patient, particularly for myosteatotic patients.

There are several limitations of this study. The limited cases of muscle atrophic patients increased the chance of false negative results. A larger cohort or multiple center studies are expected to reduce the possibility of type II error. There is a potential selection bias. Frailest patients may have been exempted from further treatment after diagnosis, e.g., based on patient wish, or that of the patient's representative. Thus, those who had too poor physical condition to undergo invasive treatment may have been excluded from this study. It is difficult to tell if this selection bias would lead to an over- or underestimation.

Also, despite the lack of association between muscle atrophy and overall survival or neurological outcome in this study, it is important to note that due to its retrospective nature CT-assessed skeletal muscle mass is solely investigated in this study. Although this approach is common in retrospective studies [9–12, 14], the terminology of sarcopenia and muscle atrophy are mixed commonly. The operational definition of sarcopenia—as defined by the European Working Group on Sarcopenia in Older People 2 (EWGSOP2)–identifies probable sarcopenia by low muscle strength first, with secondary confirmation by the presence of low muscle quantity on imaging [7]. Thus, this study included myosteatosis as an amendment to the comprehensive EWGSOP2 definition. Although alike muscle atrophy no association between myosteatosis and poor outcome was found, patients with myosteatosis are two times more likely to experience severe neurological deficiency after the aneurysmal rupture than patients without myosteatosis. This might explain why in univariable analysis, myosteatosis is a risk factor for reduced survival rate and poor neurological outcome, but such effect vanished when adjusted for WFNS and other risk factors.

Another limitation of this study is that for CT based measurements of muscle atrophy and myosteatosis, there are no standardized thresholds. In this study we have therefore performed optimal stratification to determine the threshold for myosteatosis and adopted a previously reported threshold for muscle atrophy, however further external validation is required to investigate whether these measures are generalizable. Furthermore, the measurements were made on both 3- and 5-mm slices resulting in varying spatial resolutions. Increased slice-thickness may have led an effect upon the accuracy of both the muscle volume and myosteatosis measurements, however this effect will be limited as the majority of scans (99%) were 3 mm.

## Conclusions

Myosteatosis was found to be associated with poor physical condition after the aneurysmal rupture. Skeletal muscle atrophy and myosteatosis were however found to be irrelevant to

overall survival and neurological outcome in the Western-European patient after aSAH. Future studies are needed to validate these finding.

## Author Contributions

**Conceptualization:** J. Marc C. Van Dijk, Reinoud P. H. Bokkers.

**Data curation:** Yuanyuan Shen, Abdallah H. A. Zaid Al-Kaylani, Carlina E. van Donkelaar, J. Marc C. Van Dijk, Reinoud P. H. Bokkers.

**Formal analysis:** Yuanyuan Shen, Stef Levolger.

**Investigation:** Yuanyuan Shen.

**Methodology:** Yuanyuan Shen, Stef Levolger, Alain R. Viddeleer, Reinoud P. H. Bokkers.

**Project administration:** Reinoud P. H. Bokkers.

**Resources:** J. Marc C. Van Dijk.

**Software:** Alain R. Viddeleer.

**Supervision:** Reinoud P. H. Bokkers.

**Validation:** Yuanyuan Shen, Stef Levolger.

**Visualization:** Yuanyuan Shen, Alain R. Viddeleer.

**Writing – original draft:** Yuanyuan Shen.

**Writing – review & editing:** Stef Levolger, Abdallah H. A. Zaid Al-Kaylani, Maarten Uyttenboogaart, Carlina E. van Donkelaar, J. Marc C. Van Dijk, Alain R. Viddeleer, Reinoud P. H. Bokkers.

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
