## [Decision Letter · Decision Letter 0]

12 Nov 2021

PONE-D-21-30009Skeletal muscle atrophy and myosteatosis are not related to long-term aneurysmal subarachnoid hemorrhage outcomePLOS ONE

Dear Dr. Bokkers,

Thank you for submitting your manuscript to PLOS ONE. After careful consideration, we feel that it has merit but does not fully meet PLOS ONE’s publication criteria as it currently stands. Therefore, we invite you to submit a revised version of the manuscript that addresses the points raised during the review process.

We look forward to receiving your revised manuscript.

Kind regards,

Ezio Lanza, M.D.

Academic Editor

PLOS ONE

Journal Requirements:

Reviewers' comments:

Reviewer's Responses to Questions

5. Review Comments to the Author

Reviewer #1: Specific comments:

- should be added more detail about the in-house developed software (SarcoMeas 0.34; UMCG, Groningen, The Netherlands) as described in the Vedder IR, Levolger S, Dierckx RAJO, et al. Effect of muscle depletion on survival in peripheral arterial occlusive disease: Quality over quantity. J Vasc Surg. 2020;72(6):2006- 2016.e1. doi:10.1016/j.jvs.2020.03.050

- The authors mention that “Both interobserver and intraobserver levels of agreement for SMA and SMD were found to be excellent in a prior study”, however there might be study-specific variations as the Vedder et al assessed patients with peripheral arterial occlusive disease while the current study assess cross-sectional muscle measurement in the cervical region. Taking into account the manual segmentation issue, the both interobserver and intraobserver agreement for SMA and SMD is needed.

- it's unclear how muscle atrophy was defined by a sex-specific cutoff value and how myosteatosis was defined by a BMI-specific cutoff value. Should be explain in the method section.

- should be added the background why these specific thresholds for skeletal muscle and for intra- and inter-muscle adipose tissue have been selected.

- As aging is associated with myosteatosis, it might be the effect of aging that the myosteatotic group in this study are more likely to be above the age of 70

- as a limitation, it is should be highlight that standardization of CT-derived diagnostic thresholds for muscle mass and myosteatosis is lucking

- table 1: what is the rational behind the variables age > 70y and BMI < 25. Should be stated in the material and methods section.

General comments

- references need to be homogenized, for example, introduction section line 51.

- typos line 77 (material and methods), line 118 (figure 1 legend)

- figure 1, color names in the figure legend not correspond to color in the figure

- table 1: is too busy to read.

Reviewer #2: In this retrospective study, authors evaluated the correlation between sarcopenia or myosteatosis and poor outcome after subarachnoid hemorrhage due to aneurysmatic rupture. They did not find statistically significant results with mortality and poor neurological function.

Overall, the manuscript is sound and clearly written; however, authors could improve the section of Methods according to the following suggestions.

- Details about the CT protocol parameters

- Could the thickness variability (1 to 5 mm) have influenced the cross-sectional muscle measurements?

- Number of investigators in charge of muscle mass measurement

---

## [Author Response · Author response to Decision Letter 0]

11 Jan 2022

A detailed point-by-point response to the reviewers has been attached as separate file.

---

## [Decision Letter · Decision Letter 1]

15 Feb 2022

Skeletal muscle atrophy and myosteatosis are not related to long-term aneurysmal subarachnoid hemorrhage outcome

PONE-D-21-30009R1

Dear Dr. Bokkers,

We’re pleased to inform you that your manuscript has been judged scientifically suitable for publication and will be formally accepted for publication once it meets all outstanding technical requirements.

Please note some very minor comments that you can implement in the final draft. 

Kind regards,

Ezio Lanza, M.D.

Academic Editor

PLOS ONE

Reviewer #2: Authors improved the readability and the quality of their manuscript, clarifying some important concepts in the Methods section.

- Table 3: highlight with bold type the significant p-values

- Slice thickness: “Of the 293 subjects, 289 (98.63%) had a slice thickness of 3 mm. Given that the slices varied only in 4 patients, we do not expect that this may have influenced the muscle measurements.” This idea should be clarified in the text.

---

## [Editor Report · Acceptance letter]

24 Feb 2022

PONE-D-21-30009R1 

Skeletal muscle atrophy and myosteatosis are not related to long-term aneurysmal subarachnoid hemorrhage outcome 

Dear Dr. Bokkers:

I'm pleased to inform you that your manuscript has been deemed suitable for publication in PLOS ONE. Congratulations! Your manuscript is now with our production department. 

Kind regards, 

on behalf of

Dr. Ezio Lanza 

Academic Editor

PLOS ONE